# Human arm endpoint-impedance in rhythmic human-robot interaction exhibits cyclic variations

Vincent Fortineau[1,2,3,4]*, Isabelle A. Siegler[2,3]*, Maria Makarov[1], Pedro Rodriguez-Ayerbe[1]

1 Laboratoire des signaux et systèmes, CNRS, CentraleSupélec, Université Paris-Saclay, Gif-sur-Yvette, France, 2 CIAMS, Université Paris-Saclay, Orsay, France, 3 CIAMS, Université d'Orléans, Orléans, France, 4 Auctus, Inria, Talence, France

* vincent.fortineau@free.fr (VF); isabelle.siegler@universite-paris-saclay.fr (IAS)

**Data Availability Statement:** All relevant data are within the paper and its Supporting Information files. In complement the data will be held in a public repository in an OFS dataset (https://osf.io/

## Abstract

Estimating the human endpoint-impedance interacting with a physical environment provides insights into goal-directed human movements during physical interactions. This work examined the endpoint-impedance of the upper limb during a hybrid ball-bouncing task with simulated haptic feedback while participants manipulated an admittance-controlled robot. Two experiments implemented a force-perturbation method to estimate the endpoint parameters of 31 participants. Experimental conditions of the ball-bouncing task were simulated in a digital environment. One experiment studied the influence of the target height, while the other explored the impedance at three cyclic phases of the rhythmic movement induced by the task. The participants' performances were analyzed and clustered to establish a potential influence of endpoint impedance on performance in the ball-bouncing task. Results showed that endpoint-impedance parameters ranged from 45 to 445 N/m, 2.2 to 17.5 Ns/m, and 227 to 893 g for the stiffness, damping, and mass, respectively. Results did not support such a critical role of endpoint impedance in performance. Nevertheless, the three endpoint-impedance parameters described significant variations throughout the arm cycle. The stiffness is linked to a quasi-linear increase, with a maximum value reached before the ball impacts. The observed damping and mass cyclic variations seemed to be caused by geometric and kinematic variations. Although this study reveals rapid and within-cycles variations of endpoint-impedance parameters, no direct relationship between endpoint-impedance values and performance levels in ball-bouncing could be found.

## Introduction

Humans can adjust their limbs' dynamic properties to achieve stable behavior in different environments while maintaining dexterous performances [1–3]. Measuring human interaction properties during various tasks is relevant to understanding the emergence of perturbed goal-oriented movements. Replicating these behaviors can aid in designing bio-inspired robotic control and offer insights into human motor control.

2fapk/?view_only=
f1b72ab1413d43178f24c510658ade3e).

**Funding:** The authors received no specific funding for this work.

**Competing interests:** The authors have declared that no competing interests exist.

Researchers have extensively studied the mechanical impedance of limbs under the control prism [4–6], as it relates kinematics to forces, and enables modeling human movements during physical interactions. However, studying mechanical impedance is strenuous since it cannot be measured directly. Estimating impedance requires perturbing the limbs and comparing the observed movement to an unperturbed trajectory. Two types of tasks can be easily distinguished: 1) static or isometric [7, 8], where the unperturbed pose is well known, and 2) dynamic, where the estimation of the unperturbed trajectory is much more complex [9, 10]. The current paper focuses on the latter with *hybrid* movements that involve intermittent interactions with the environment. The choice of modeling type is crucial for understanding.

To delve into the neural processes governing dynamic limb control, comprehensive models are indispensable [11–14]. To account for passive muscle viscoelastic properties and active behaviors induced by reflexes and motion, Latash and Zatsiorsky [14] explained that a single joint would at least need 13 parameters. They considered two delays for the viscoelastic contribution of both the short-latency monosynaptic reaction and the tonic stretch reflex, with their respective stiffness and damping. Extending this analysis to multiple joints in movement becomes exceedingly complex and practically unattainable with such advanced models. Despite their limitations [15], elementary models, such as mass (M)—spring (K)–damper (B) models (KBM or KBI) or Hill-type muscle models, have been shown to approach human behavior [3] and performances [16]. These models do not intend to explain the complete human motor control but rather reproduce some of its aspects. These simple models have been widely studied at the endpoint level [1, 3, 7, 17], notably concerning impedance during multi-joint movements [1, 17]. By decoupling these basic models from biological considerations and adopting a behavioral approach, researchers have successfully analyzed performance and expertise [17], learning [1, 16], and exploration [18].

The study of impedance variations during movements and particularly cyclic movements with contact phases and hybrid transitions is especially relevant for locomotion. Previous research has thoroughly investigated time-varying stiffness in the ankle joint during walking [19] or running [20], as well as the contributions of reflexes and intrinsic viscoelastic properties to this stiffness [11]. Huang and Wang [21] designed a Central Pattern Generator (CPG)-based human locomotion model, incorporating control over both joint torque and joint stiffness (stiffness constant and equilibrium point). They improved the disturbance rejection on uneven terrain compared to previous models. The time-varying viscoelastic properties of the upper limbs have also been investigated in free cyclic movements [22] and rhythmic movements under various loads [23]. Lacquaniti et al. [24] and Tsuji and Tanaka [3] studied the endpoint impedance of a ball-catching hybrid task (with both free movements and contact phases) with predictable physical interactions.

The present study characterizes the upper-limb mechanical impedance in a hybrid rhythmic task, such as a ball-bouncing task with haptic feedback. The ball-bouncing task has already been studied for its apparent simplicity to address several motor control issues [25–29]. Avrin et al. [30] implemented a CPG-based control structure to reproduce human-like kinematic performances in the ball bouncing. However, in the experimental data used to validate this model [27, 28], the dynamics of the physical interactions between the ball, the paddle, and the human were not simulated. A new experimental setup designed in a previous experiment [31] overcame this shortcoming using an admittance-controlled robot that participants manipulated to vertically move a virtual paddle and bounce a virtual ball in a digital environment. The inclusion of haptic feedback during ball-paddle impacts is particularly relevant as it enhances performance by reducing noise in ball state estimation [25]. Later studies [32, 33] validated the methodology for endpoint-impedance estimation during cyclic movements. In this study, two

experiments evaluate endpoint impedance in an ecologically valid multi-joint rhythmic movement with low constraints imposed by the robot.

The primary goals of this research are threefold: 1) to investigate whether endpoint impedance parameters exhibit variations during a cyclic task and, if so, to discern their nature; 2) to examine whether there is a correlation between endpoint impedance behavior and the task's constraints; and 3) to explore potential links between endpoint impedance parameters and the level of performance achieved, as suggested by previous research on this topic [17]. The results presented provide valuable insights into these questions, shedding light on the nature of endpoint impedance during complex rhythmic tasks.

## Materials and methods

### Participants

Thirty-one healthy participants with no known motor disorders took part in the experiment. Among the participants, 25% were women, 10% were left-handed, 40% had already manipulated a similar robotic system, and 30% had never interacted physically with a robot. On average (SD standard deviation), participants were 30.0 years old (SD 8.6), declared a height of 176.2 cm (SD 9.3), and weighed 71.3 kg (SD 12.7). Before the experiment, participants were required to read and sign a written informed consent form approved by the Ethics Committee for Research (CER) of the Université Paris-Saclay (file number 217, September 2020). The Ethics Committee also approved the experimental procedure.

### Apparatus

Fig 1 depicts the experimental setup used for the experiment. Participants stood in front of a table supporting a KUKA youBot 5 DOF robot. The robot had its first and last joints position-controlled to a fixed orientation to evolve only in the participants' sagittal plane. The other three joints were admittance-controlled at 1 kHz, as detailed in Fortineau et al. [31], so that participants could move the robot along a vertical line without feeling the robot's weight, joints' friction, and with reduced apparent inertia [34]. Forces of interaction were measured using a 6-axis force/torque sensor ATI Mini 45, placed between the robot endpoint and a handle that participants maneuvered. The data from the sensor was sampled at 1 kHz. For the control, the endpoint position of the robot was obtained using joint data. However, to limit bias from joint flexibility, a motion capture system V120:Trio by Optitrack measured handle position with a submillimetric accuracy at 120 Hz. This position measurement was used for data analysis. ROS was used on a master computer for the control, and the environment was simulated in one dimension with ROS and Rviz.

### Ball-bouncing task

Participants had to operate the robot handle in order to move a paddle in a digital environment where the ball-bouncing task was simulated. The ballistic equation determined the ball kinematics during free movement, and the ball post-impact velocity used a restitution coefficient $\alpha = 0.6$ to account for energy loss, as described in Fortineau et al. [31]. The simulation was calibrated so that the participants could approach performances close to the ones described in [27, 28] to guarantee rhythmic movements. The uni-dimensional task was displayed on a screen, with the ball represented by a sphere (mass: 0.058 kg, radius: 0.0325 m) and the paddle by a bar (mass: 1 kg).

The goal of the task was to match the ball's apex position with a target height $h^*$, kept constant during each trial and materialized by a horizontal line. A score was displayed slightly

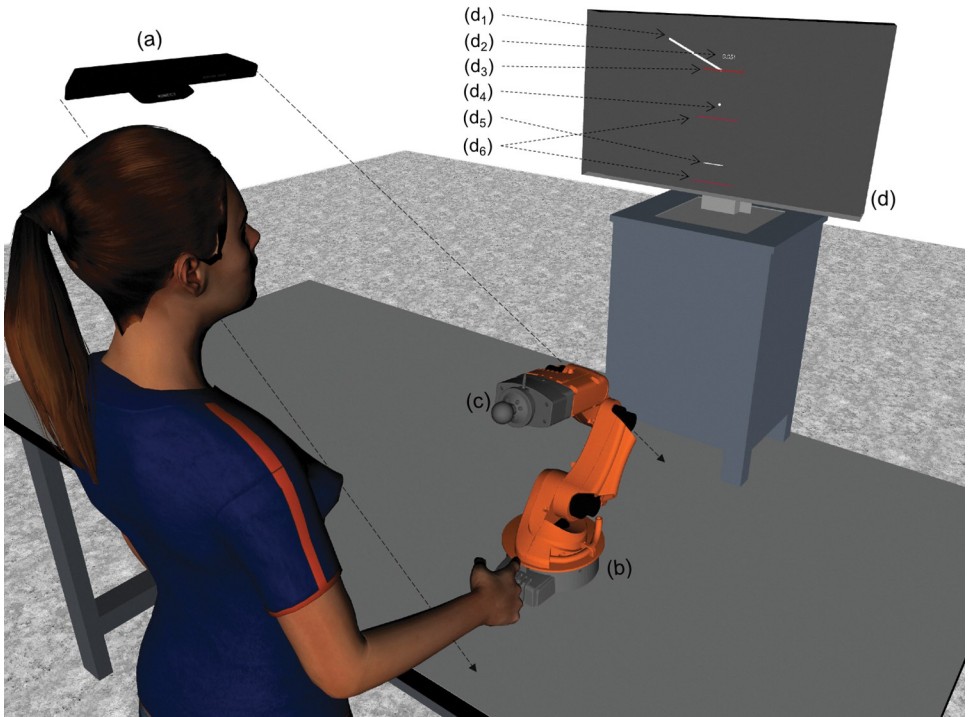

**Fig 1. Experimental setup.** Behind the robot, a screen (d) displays the simulated environment on an elevated surface so that the monitor is roughly at the eye level of the participants. The motion capture system (a) is positioned on a separate surface facing the robot (b). A force-torque sensor is positioned between the endpoint of the robot and the handle (c). The screen shows: (d1) a ramp, (d2) a score, that is the distance between the ball center and the target height (d3), a ball (d4), a paddle (d5), and the limits for the robot control (d6).

above the target line to help participants reduce the bouncing error $\varepsilon_r$ between the ball center at the apex and $h^*$. Participants were instructed to maintain a constant ball impact position $z^k$ during trials. During both experiences, the robot introduced perturbations timed from the ball impacts on the simulated paddle at three possible cyclic phases: ($\varphi_1$) after the ball impact, ($\varphi_2$) during the decreasing phase of the paddle, and ($\varphi_3$) before the subsequent ball impact near the lowest position. These phases are shown in Fig 2. Perturbations were randomly spaced between 2 and 5 s.

**Haptic feedback & perturbations.** The robot admittance control loop was interrupted for 30 ms, and a force along the vertical axis was introduced using only the torque control loop, with a constant setpoint to simulate ball impacts. The Eq (1) adapted from Kawazoe [35] determined the magnitude of the force $f_i^k$, with $b$ and $p$ indicating either the ball or the paddle, $v^{k^-}$ the velocity right before the impact $k$, $m$ the mass, $\hat{K} = 650$ an estimated coupled stiffness between the arm and the ball, and $\alpha$ the same restitution coefficient used for the ballistic kinematics.

$$f_i^k = -\left(v_b^{k^-} - v_p^{k^-}\right)(1+\alpha)\frac{\sqrt{m_b \hat{K}}}{\pi\sqrt{1+\frac{m_b}{m_p}}} \tag{1}$$

Perturbations were introduced through the same pathway in stochastic directions. Fig 3 shows an example of two resulting endpoint force perturbations. The peak value is reached

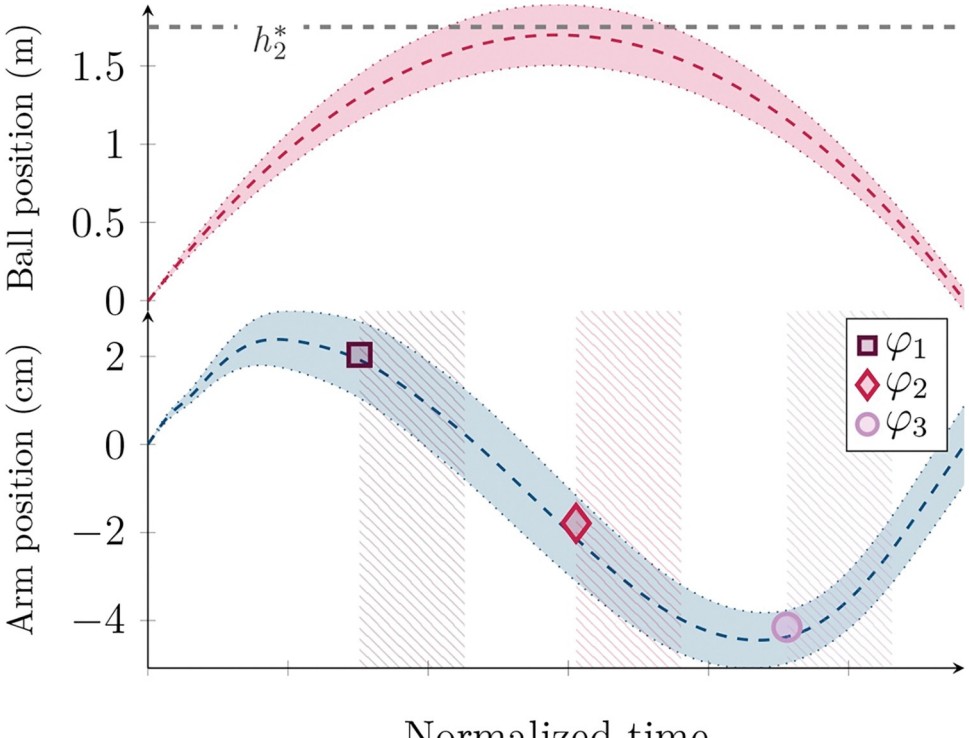

**Fig 2. Phases perturbed along a time-normalized cyclic position trajectory.** The colored area around the trajectories represents the standard deviation. The hatched area represents the amount of data used for impedance identification. The ball trajectory axis is provided in meters, while in centimeters for the hand trajectory.

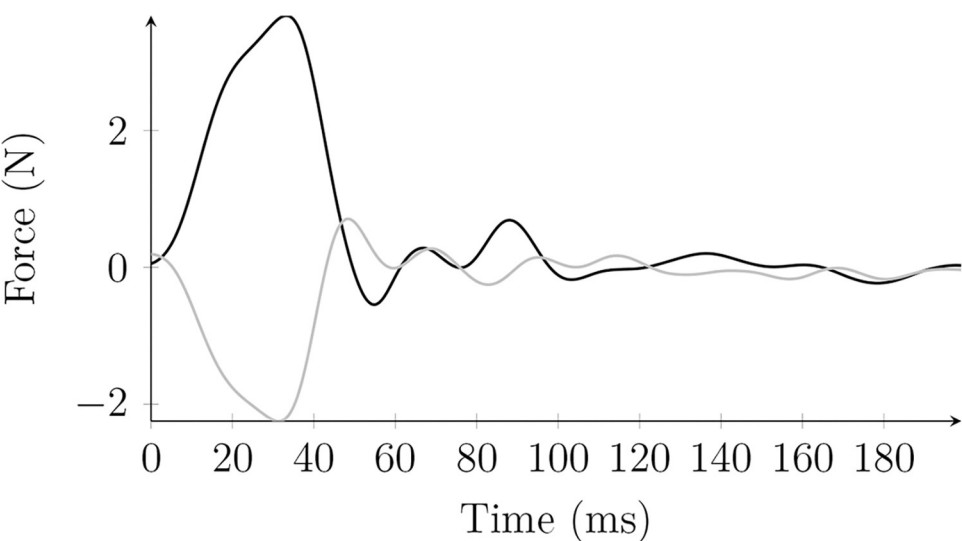

**Fig 3. Typical force perturbations.** Two examples of force perturbations resulting from the introduction of torque-controlled perturbations.

approximately 30 ms after the onset of the torque control, and the next front lasts less than 30 ms afterward. The total duration of a perturbation is, therefore, below 60 ms.

## Mechanical impedance modeling

Eq (2) shows the well-known KBM model used in this study.

$$\delta f = K\delta x + B\delta \dot{x} + M\delta \ddot{x} \tag{2}$$

This model relates a differential force $\delta f$ to a differential position $\delta x$ when a perturbation force (equal to $\delta f$) is introduced along the virtual trajectories $f_0$ and $x_0$. The differentials are the differences between virtual and perturbed trajectories (measured). The model can be seen as a perturbation rejection system with the linear parameters $K$, $B$, and $M$: the stiffness, damping, and mass, respectively. Virtual trajectories [36] refer to completely unperturbed behaviors and are approximated using the nominal trajectories without the perturbations described in the previous section. The model uses its decoupled form, which was already applied by Erden and Billard [17, 37]. This work only studies the impedance along the movement.

The virtual paddle position was approximated using spline interpolation with starting and landing points, respectively, right before the perturbation and 350 ms after. This window of 350 ms was chosen to minimize the quadratic error with an undistorted trajectory and maximize the duration to account for the arm's time response. The virtual force was estimated using a sum of three sines optimized on portions of the signal around a masked window of 100 ms after the perturbation [33]. The sine model reached a coefficient of determination of 95.1% (SD 5.3) for reconstructing the virtual force of nonperturbed trajectories on the duration of the masked interval [33]. Fig 4 shows an example for both trajectories.

The impedance parameters were first estimated using an ARX (Auto Regressive model with eXternal inputs) least square identification. The estimation on a discretized impedance model (3), with $q^{-1}$ the delay operator and $e$ white noise, does not rely on the derivatives of the differential position [33]. The relationships between the discrete-time coefficients ($a_i$ and $b_i$) and the continuous-time impedance parameters $K$, $B$, and $M$ are given in (4). They can be obtained with a zero-order hold discretization with a time step of $\Delta t$ equal to the sampling time of 1 ms. An identification window of 150 ms was chosen to be minimal while still allowing proper identification [33]. Automatic responses or voluntary actions can occur with 100 to 180 ms delays [38, 39] and cannot be modeled with a linear impedance model. They were, however, considered not dominant even after 200 ms [17, 37].

$$\delta x[t_k] = (b_1 + b_0 q^{-1})\delta f[t_k] - (a_1 q^{-1} + a_0 q^{-2})\delta x[t_k] + e(t_k) \tag{3}$$

$$\begin{cases} K = \dfrac{1 + a_1 + a_0}{b_1 + b_2} \\[2mm] B = -\dfrac{\ln(a_0)M}{\Delta t} \\[2mm] M = -\dfrac{K\Delta t^2}{\ln(0.5(-a_1 - \sqrt{a_1^2 - 4a_0}))(\ln(0.5(-a_1 - \sqrt{a_1^2 - 4a_0})) - \ln(a_0))} \end{cases} \tag{4}$$

## Experimental protocol

Two experiments (Experiment P, for phase, and Experiment H, for height) occurred during two sessions on the same day (for all but three participants), separated by at least 45 min. Each

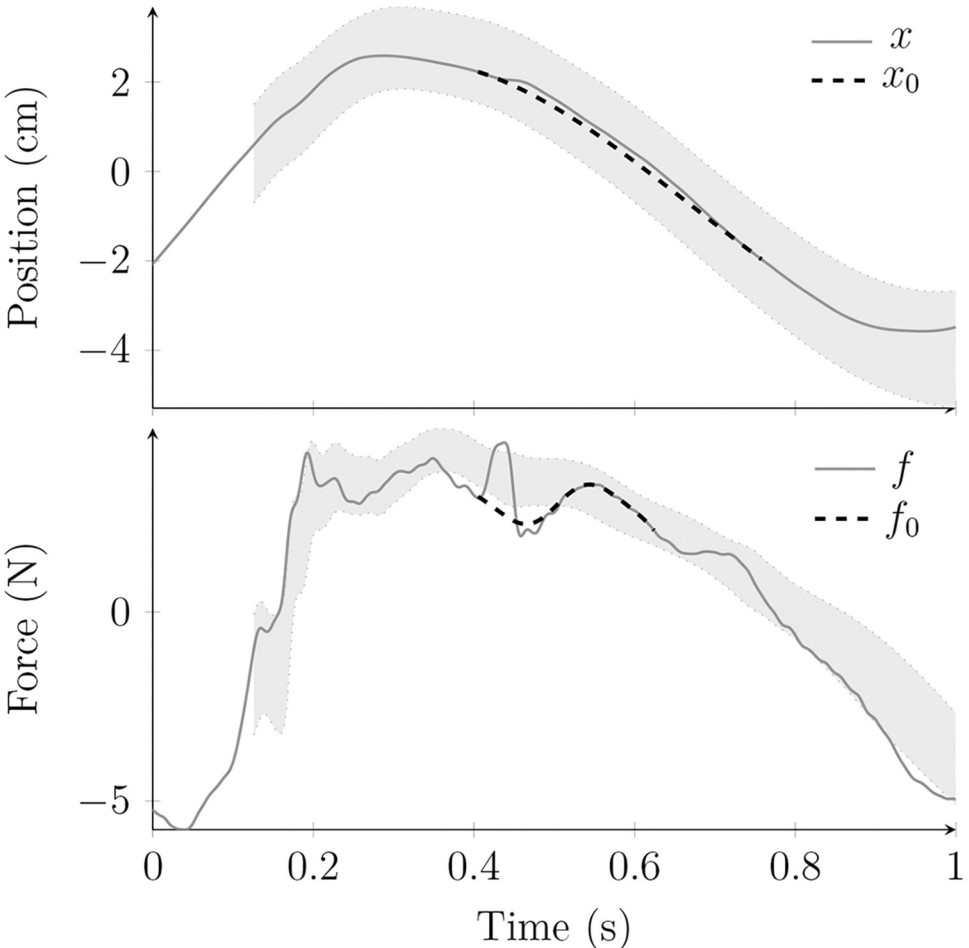

**Fig 4. Position and force cyclic trajectories of a participant.** Position (x) and force trajectories (f) are in solid lines, with the estimated virtual trajectories in dashed lines, for a single cycle. The grey areas represent the standard deviation of the trajectories recorded for this participant.

session lasted 30 min on average. The first session (Session 1) consisted of a familiarization phase and then two trials of 5 min each with a brief pause in between. Participants started the first trial when they were confident enough with the ball-bouncing task. The second session (Session 2) consisted of three other trials of 5 min each, also separated by small pauses to limit fatigue.

Experiment P consisted of three trials: the two trials of Session 1 and the last trial of Session 2, or the three trials of Session 2. The target height $h_2^*$ was fixed at 1.75m, and the three cyclic phases $\varphi$ were randomly perturbated in all trials.

Experiment H consisted of two trials (the two trials of Session 1 or the two first trials of Session 2). The two trials randomly attributed a target height of either 1.5 m ($h_1^*$) or 2 m ($h_3^*$). In this experiment, all perturbations occurred at phase $\varphi_1$.

Participants were randomly selected to start with Experiment H (17 participants) or Experiment P (14 participants), which divided the participants into two experiment-order subgroups. After each experiment, participants were asked to report their perception of haptic feedback and perturbations.

## Data analysis

With 31 participants, 155 trials led to more than 14 hours of recorded data, not including the familiarization phase. All acquired data was interpolated on a time vector sampled at 1 kHz. The position and force were filtered using Butterworth second-order digital filters with cutoff frequencies of 25 Hz and 50 Hz, respectively. Cycles were defined using ball impacts, and the distance between the target height and the ball's apex defined the bouncing error for each cycle. The data from the simulated environment was not used for the impedance estimation.

We chose to distinguish three performance levels in both experiments' data, defined as novice $e_1$, intermediate $e_2$, and advanced $e_3$. The individual performance was first evaluated using the interquartile range of the bouncing errors $\varepsilon_r$ on each participant's data, and the median of the quadratic bouncing errors $\sqrt{\varepsilon_r^2}$ as the first measures of repeatability. These groups were generated using a k-means clustering algorithm on the interquartile range of $\varepsilon_r$ since repeatability was proven to qualify performances [40].

For the rest of the analyses on performance, two dependent variables were used to characterize the task, using the same method as previous studies on ball bouncing. The first variable defined the precision with the average bouncing error $\bar{\varepsilon}_r$. The second variable defined the repeatability using the standard deviation of the bouncing errors $(\varepsilon_r)$.

On average, 32.4 (SD 6.1) impedance parameter identifications for each experiment, configuration ($h^*$ and $\varphi$), and participant were used to estimate the endpoint-impedance parameters. The estimated parameters for each configuration and participant were obtained using the median of these identifications, since they proved to be more accurate than average values after taking out outliers. This was observed with a simulated impedance model on experimental signals. Each impedance parameter was populated with 93 estimations (31 participants × 3 phases) for Experiment P and with 62 estimations for Experiment H (31 participants × 2 target heights).

The adequation of the Impedance parameters of each identification was evaluated using the coefficient of determination $R^2$ of the reconstruction of the perturbed position trajectory with the identified parameters and the differential force trajectory.

We used JASP (version 0.16) to perform the analyses of variance (ANOVA), considering a threshold of 5%. We also conducted post-hoc tests using the Bonferroni correction. Only significant results are detailed in the result section. We used Matlab (release 2020b) to process data.

## Results

### Effect of perturbations on participants' performance

Perturbations were programmed to be as seamless as possible while providing sufficient deviations for impedance estimations [33]. Even if participants were warned of both haptic feedback and perturbations, only 55% declared having felt perturbations during at least one of the trials.

All the bouncing errors taking place the cycle before a perturbation $c^{-1}$ and three cycles after ($c^1$, $c^2$, $c^3$) were averaged for each participant and compared using a mixed-design ANOVA (4 cycles × 2 experiments) to assess the effect of perturbations on performance. Only a significant effect of cycle order on $\bar{\varepsilon}_r$ was observed $F(3, 180) = 6.8$, $p < 0.001$, $\eta^2 = 0.026$. A post-hoc analysis revealed that the bouncing error at $c^1$ was significantly larger than other cycles' errors, with a mean difference of 2.0 cm (Fig 5).

A second mixed-design ANOVA conducted on $\sigma(\varepsilon_r)$ with the same factors had to be corrected [41] since the sphericity test was not verified [42]. No significant effect was observed for the main

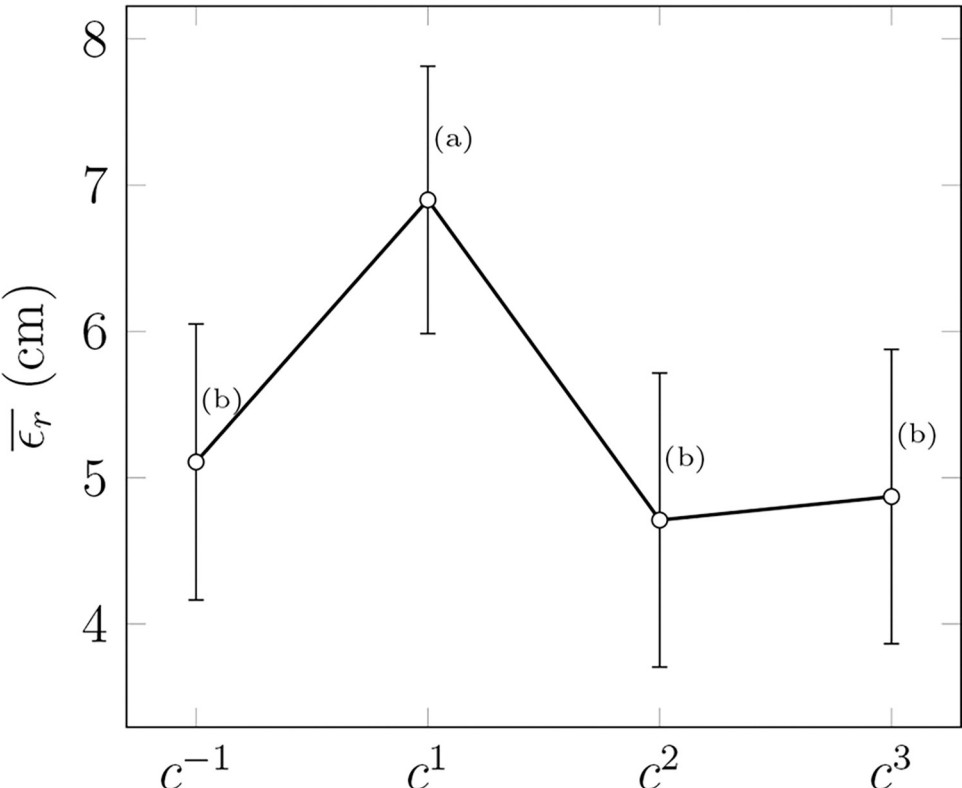

**Fig 5. Mean bouncing error $\bar{\varepsilon}_r$.** The errors are given as a function of cycles around a perturbation ($c^{-1}$ being right before the perturbation and $c^1$ right after), with standard errors used as confidence intervals. Groups that are not significantly different are marked with the same letter for each target height.

effect factor, the random effect, or the interaction. Therefore, the repeatability of the bouncing error was similar for all cycles, with an average standard deviation of 25.1 cm (SD 5.0).

## Performance-based clustering of participants

Among all participants, 40% were already familiar with the experimental setup, while 50% had never manipulated an articulated robot. K-means analysis showed that one participant had to be discarded from the computation of group performances because their score was too low, thus putting the participant in their group. The clustering was populated with 5 advanced, 13 intermediate, and 13 novice users. Interestingly, a similar clustering was obtained using the quadratic errors. Only 3 participants were classified differently.

To validate the essence of the clustering, the differences between performance-level groups $e_i$ were studied using two one-way ANOVAs, on the precision using the mean bouncing error $\bar{\varepsilon}_r$ and on the repeatability using the standard deviation $\sigma(\varepsilon_r)$. The ANOVA on $\bar{\varepsilon}_r$ unveiled no significant effect of the level-group factor $F(2, 28) = 3.0, p = 0.064, \eta^2 = 0.178$, while the ANOVA on $\sigma(\varepsilon_r)$ revealed a strong and significant effect of the performance-level factor $F(2, 28) = 33.0, p < 0.001, \eta^2 = 0.702$.

The paddle acceleration at impact [29, 43] offers another way of looking at individual performances. However, the accelerations estimated with two-points numerical derivatives were not significant between the performance-level groups. Participants hit the ball with an average post-impact acceleration of 0.29 ms$^{-2}$ (SD 1.69).

## Experiment H

**Performances.**   For target heights $h_1^* = 1.5$ m and $h_3^* = 2$ m, the mean ball-bouncing magnitudes (difference between the apex and impact position) were 1.176 m (SD 0.019) and 1.670 m (SD 0.019), respectively. These values show that participants hit the ball at similar vertical positions in both conditions (0.324 m for $h_1^*$ and 0.330 m for $h_3^*$) and did not tend to hit it at a higher position when the target was higher.

The effect of target height $h_i^*$ and performance-level groups $e_i$ on $\bar{\varepsilon}_r$ was analyzed using a mixed-design ANOVA (between: 2 × within: 3), revealing a significant main effect of factor $h_i^*$, $F(1, 28) = 7.0$, $p = 0.014$, $\eta^2 = 0.043$. Neither the effect of performance level $e_i$ nor the interaction was significant. Overall, the mean errors for $h_1^*$ and $h_3^*$ were 6.8 cm (SD 6.3) and 3.2 cm (SD 7.1), respectively.

On the whole group, the mean $\sigma(\varepsilon_r)$ increased from 21.7 cm (SD 4.2) to 27.2 cm (SD 6.1) for $h_1^*$ and $h_3^*$, respectively. A mixed ANOVA (between: 2 $h_i^*$ × within: 3 $e_i$) revealed a significant main effect of $h_i^*$, $F(1, 28) = 44.8$, $p < 0.001$, $\eta^2 = 0.179$, a significant effect of $e_i$, $F(1, 28) = 19.6$, $p < 0.001$, $\eta^2 = 0.404$, but no significant interaction. The post-hoc analysis revealed that the level of performance was significantly different for each pair. Fig 6 details the variations of the mean $\sigma(\varepsilon_r)$ according to both factors.

**Endpoint-impedance parameters.**   The same factors $h_i^*$ and $e_i$ were analyzed for each endpoint-impedance parameter. ANOVAs did not yield any significant results for the stiffness nor the damping. The average estimated stiffness was 116 N/m (SD 60), with a minimum value of 45 N/m and a maximum of 401 N/m. The average estimated damping was 9.9 Ns/m (SD 3.1), with a minimum value of 2.2 Ns/m and a maximum of 16.8 Ns/m.

The mixed ANOVA about the estimated mass revealed a significant effect of the main factor $h_i^*$, $F(1, 28) = 4.6$, $p = 0.040$, $\eta^2 = 0.014$. Neither the effect of performances $e_i$ nor the interaction was significant. The average mass estimated for $h_1^*$ was 459 g (SD 126) and 434 g (SD 94) for $h_2^*$ with a minimum value of 272 g and a maximum of 893 g.

## Experiment P

**Performances.**   The one-way ANOVA for each variable used to characterize performances was conducted on the regrouped data of the three trials, studying the effect of performance clusters.

The ANOVA about the average bouncing error $\bar{\varepsilon}_r$ revealed no significant effect of the performance clusters $e_i$. The mean error for $h_2^*$ (Experience P) was 5.4 cm (SD 5.1). The ANOVA about the standard deviation of the error $\sigma(\varepsilon_r)$ revealed a significant effect on the performance clusters $e_i$, $F(2, 28) = 24.0$, $p < 0.001$, $\eta^2 = 0.631$. The average standard deviation was 27.8 cm (SD 3.6), 22.7 cm (SD 2.1), and 18.2 cm (SD 2.0), respectively, for the clusters $e_1$, $e_2$, and $e_3$. The post-hoc analysis showed that each pair had significant differences. Fig 7 details the mean variations of $\sigma(\varepsilon_r)$ and $\bar{\varepsilon}_r$ according to the performance clusters.

**Endpoint-impedance parameters (cyclic phase factor).**   The factors $\varphi_i$ and $e_i$ were analyzed for each endpoint-impedance parameter estimated using mixed-design ANOVAs (between: 3 × within: 3).

The mixed ANOVA about the endpoint stiffness revealed a significant effect of $\varphi_i$, $F(2, 56) = 8.3$, $p < 0.001$, $\eta^2 = 0.134$. Neither the effect of performances $e_i$ nor the interaction was significant. The post-hoc analysis indicated that the variations were significant for the couple ($\varphi_1$, $\varphi_3$). The stiffness progressively increased from $\varphi_1$ 125 N/m (SD 78) to $\varphi_3$ 191 N/m (SD 64), as shown in Fig 8i. The stiffness values were bounded between 53 N/m and 445 N/m.

The mixed ANOVA about the endpoint damping revealed a significant effect of $\varphi_i$, $F(2, 56) = 8.6$, $p < 0.001$, $\eta^2 = 0.138$, and of performances $e_i$ $F(2, 56) = 6.4$, $p = 0.005$, $\eta^2 = 0.123$. No

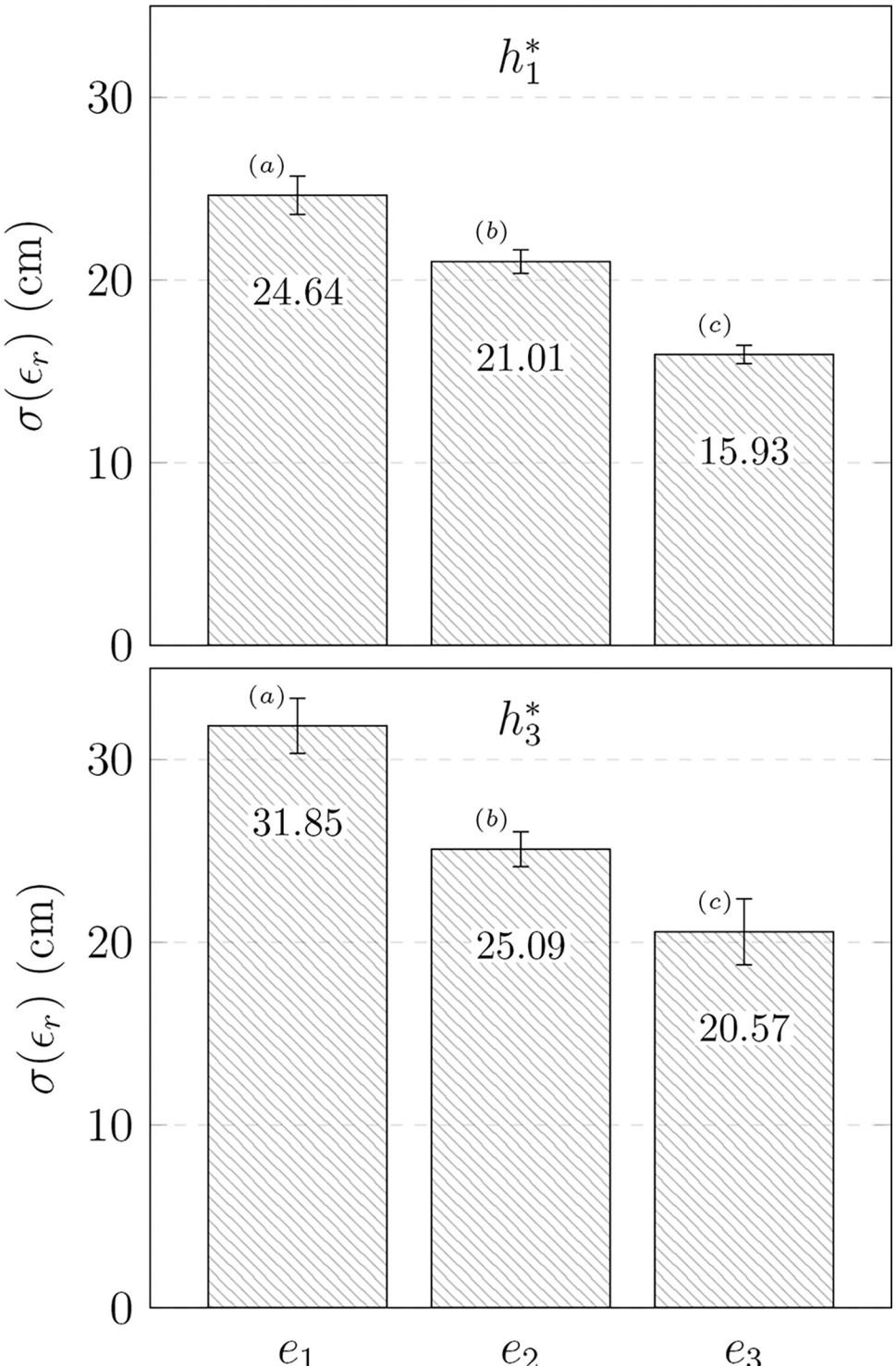

**Fig 6. Mean repeatability $\sigma(\varepsilon_r)$ of experiment H.** The repeatability is given according to performance group levels ($e_i$) and target heights ($h^*_i$), with standard errors used as confidence intervals. Groups that are not significantly different are marked with the same letter for each target height.

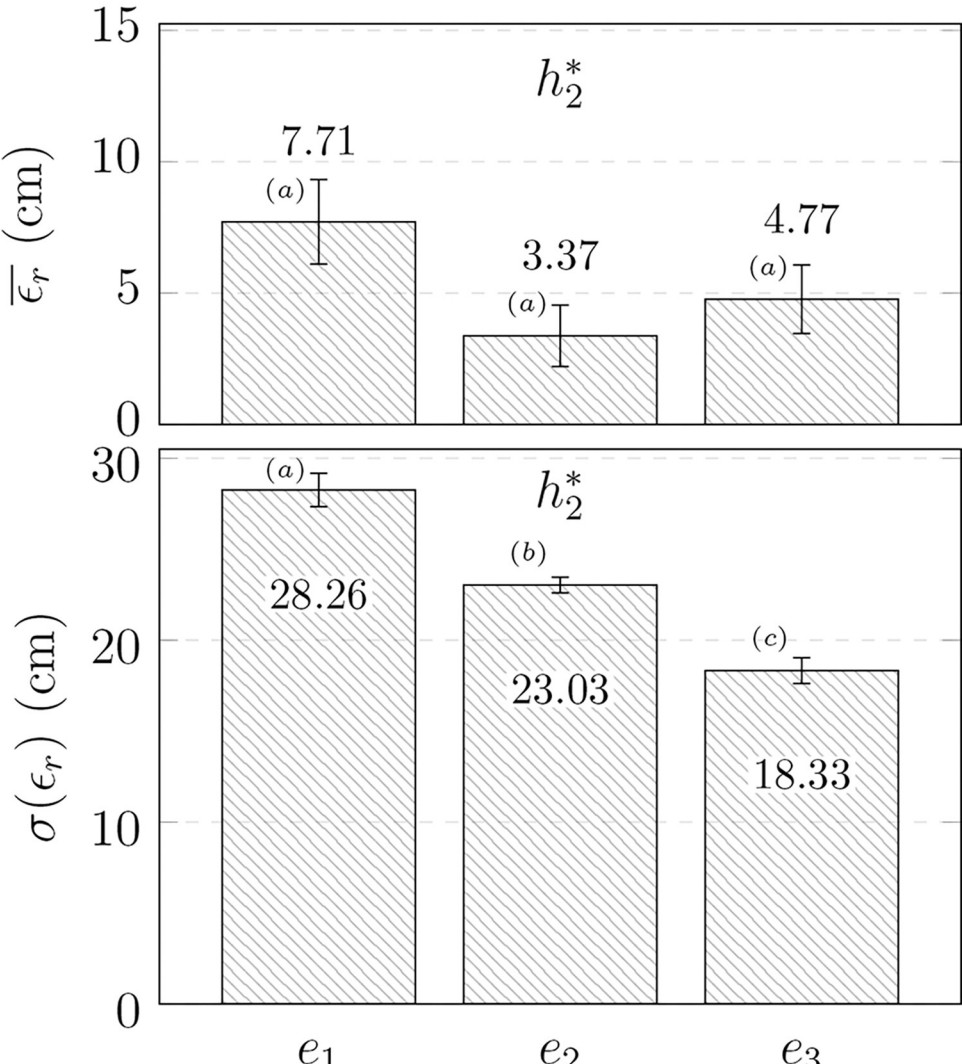

**Fig 7. Mean precision $\bar{\varepsilon}_r^-$ and repeatability $\sigma(\varepsilon_r)$ of experiment P.** The data of experiment P (target height $h_2^*$) is provided as a function of performance clusters ($e_i$), with standard errors used as confidence intervals. Groups that are not significantly different are marked with the same letter for each ANOVA.

significant interaction was observed. The post-hoc analysis indicated that the variations were significant for the couples ($\varphi_1$, $\varphi_2$) and ($\varphi_2$, $\varphi_3$), with an average difference of 1.9 Ns/m and 2.2 Ns/m, respectively. Moreover, the clusters of performances were significantly different for the couples ($e_1$, $e_2$) and ($e_2$, $e_3$). All damping values were bounded between 3.5 Ns/m and 17.5 Ns/m. The cyclic behavior of the damping is shown in Fig 8ii.

The mixed ANOVA about the apparent mass used Huynh-Feldt's (1976) correction because the sphericity assumption was violated ($p < 0.05$). It revealed a significant effect of the main factor $\varphi_i$, $F(1.5, 41.3) = 16.2$, $p < 0.001$, $\eta^2 = 0.063$. Neither the effect of performances $e_i$ nor the interaction was significant. The post-hoc analysis indicated that the variations were significant for the couples ($\varphi_1$, $\varphi_2$), with an average difference of 71 g, and ($\varphi_2$, $\varphi_3$), with an average difference of 67 g. All mass values were bounded between 227 g and 815 g. The cyclic behavior of the mass is shown in Fig 8iii.

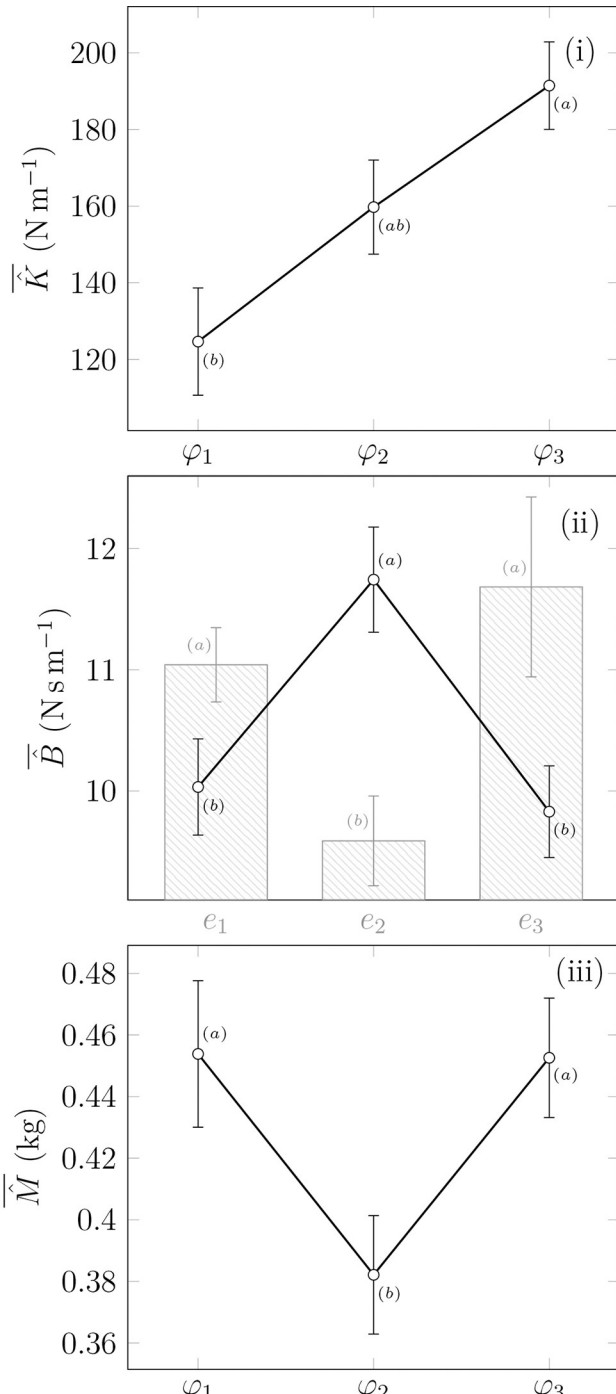

**Fig 8. Mean estimated impedance parameters ($K$, $B$, and $M$), according to the phase perturbed (and expertise in the histogram when significant differences were observed), with standard errors used as confidence intervals.** Groups that are not significantly different are marked with the same letter for each ANOVA.

The mixed ANOVA about the coefficients of determination used Huynh-Feld's correction since the sphericity assumption was violated ($p < 0.05$). It revealed a significant effect of the main factor $\varphi_I$, $F(1.6, 45.9) = 31.1$, $p < 0.001$, $\eta^2 = 0.302$. Neither the effect of performances $e_i$

nor the interaction was significant. The post-hoc analysis indicated that the variations were significant for the couples ($\varphi_1$, $\varphi_2$), with an average difference of 3.3%, and ($\varphi_1$, $\varphi_3$), with an average difference of 2.6%. Phase $\varphi_1$ had the lowest average score, with 93.5% (+/-2.7%). The minimum users' coefficient of determination (88.9%) was obtained in Phase 1 and the maximum (98.6%) in Phase 3.

The factors $\varphi_i$ and $e_i$ were analyzed for each endpoint-impedance parameter notch interval (median 95% confidence interval) using mixed-design ANOVA (between: 3 × within: 3).

The mixed ANOVA about the endpoint stiffness variability used Huynh-Feld's correction since the sphericity assumption was violated ($p<0.05$), revealing a significant effect of $\varphi_i$, $F(1.4, 39.0) = 4.8$, $p = 0.024$, $\eta^2 = 0.094$. Neither the effect of performances $e_i$ nor the interaction was significant. The post-hoc analysis indicated that the variations were significant for the couple ($\varphi_1$, $\varphi_2$), with a decrease from 89 N/m (SD 58) to 52 N/m (SD 20). The phase $\varphi_3$ had an average notch of 78 N/m (SD 32).

The mixed ANOVA about the endpoint damping variability revealed a significant effect of $\varphi_i$, $F(2, 56) = 3.5$, $p = 0.037$, $\eta^2 = 0.041$, and the interaction $F(4, 56) = 2.7$, $p = 0.037$, $\eta^2 = 0.065$. However, the effect of performances $e_i$ is not significant. The post-hoc analysis indicated that the variations were significant for the couple ($\varphi_1$, $\varphi_2$), with a decrease of 0.5 Ns/m (SE 0.2). The damping notches ranged from 1.1 Ns/m to 4.8 Ns/m.

The mixed ANOVA about the endpoint mass variability used Huynh-Feld's correction since the sphericity assumption was violated ($p < 0.05$). It revealed a significant effect of $\varphi_i$, $F(1.4, 39.0) = 8.0$, $p = 0.004$, $\eta^2 = 0.114$. Neither the effect of performances $e_i$ nor the interaction was significant. The post-hoc analysis indicated that the variations were significant with phase $\varphi_2$ and both other phases, with a lower notch value of 32 g (SD 16), compared to 56 g (SD 33) and 55 g (SD 25) for $\varphi_1$ and $\varphi_3$, respectively.

## Discussion

This work focuses on the endpoint-impedance of the upper limb during a ball-bouncing task with haptic feedback. The results of the experiments offer insights into the main topics raised: 1) analyzing the endpoint-impedance variations during a cyclic task, 2) looking for links between the constraints of the environment and the endpoint-impedance behavior, and 3) looking for a relation between performance and the endpoint impedance. Results are compared with the literature, and the perturbations' influence is discussed to comfort the methodology.

### Impedance estimation validation during a ball-bouncing task

As presented in the section "Effect of perturbations on participants' performance", only the cycles after a perturbation unveiled a significant mean error $\bar{\varepsilon}_r$ increase of about 2.0 cm on average. This increase can be considered negligible with respect to the mean bouncing magnitude of 1.42 m and the standard deviation of the bouncing error $\sigma(\varepsilon_r)$, around 25 cm for all participants. Moreover, the ANOVA on $\sigma(\varepsilon_r)$ did not underline significant differences between the cycles, showing that their repeatability did not deteriorate. Therefore, the methodology used to introduce these perturbations is close to the main objective of total transparency or seamlessness, which could allow online impedance estimation without disturbing the studied task.

The endpoint-impedance parameters estimated ranged from 45 to 445 N/m, 2.2 to 17.5 Ns/m, and 227 to 893 g for the endpoint stiffness, damping, and mass, respectively. No constraints were imposed on the participants, thus inducing the involvement of at least three joints (wrist, elbow, shoulder) to whom the hip joint might be added. A projection of 52 (13 × 4) parameters

(without considering coupling effects between joints) would have been required for joint impedance analysis. With only three parameters, 94.6% (SD 2.9) of the observed behavior could be reproduced on average for the case of slight deviations from trajectory imposed by a rhythmic task, pointing out the adequacy of the linear impedance model used.

Moreover, the endpoint parameters obtained from the linear model are comparable to those found in the literature. For point-to-point movements, Burdet et al. [1] found higher stiffness parameters ranging from 150 to 700 N/m. In the preparation of ball impacts, Tsuji and Tanaka [3] found stiffness from 44 to 189 N/m, and in a collaborative welding task, Erden and Billard [17] found similar stiffness values from 49 to 339 N/m in the direction of the movement. This study's apparent endpoint damping and mass are slightly lower than Tsuji and Tanaka's [3] and Erden and Billard's [17] observations. Lower damping values might be explained by the kinematic requirements of the ball-bouncing task and lower masses by the geometric configuration imposed by the experimental setup.

### How can the environment or task constraints influence the impedance?

Experiment H revealed a significant effect of target height on participant's performances. As the bouncing magnitude increased, their repeatability (standard deviation error) deteriorated, but their precision (mean bouncing error) slightly improved. The statistical effect was stronger for repeatability than for precision. The change in target height induces variations in kinematics, notably in frequency and ball post-impact velocity. These factors alter the constraints necessary to accomplish the task, which may explain the observed changes in performance.

Regarding impedance parameters, the only significant effect observed was related to the participant's apparent mass. However, the effect was small, with an average reduction of 25g when the height was increased by 0.5m (in the simulation). This reduction should be compared with the standard deviation of the participant mass for each estimation, which was approximately four times higher. These results may indicate either insignificant dynamic changes induced by the environmental modification, or, as detailed later, that impedance is not crucial for performing this task.

### Are there cyclic variations of the endpoint-impedance parameters?

In the preparation of a predictable contact with an environment, joint stiffness in the case of the ankle for locomotion [19] or elbow and wrist in a ball-catching task [24] increased before the contact. Tsuji and Tanaka [3] also found an increase in stiffness at the endpoint level right before motion compared to a stable posture. They deduced that the impedance regulation occurred before starting motions for a task. However, shreds of evidence could imply neurophysiological differences in the emergence of rhythmic movement in opposition to discrete or point-to-point movements [44, 45], even if unified theories have been proposed [46, 47]. Consequently, impedance variations in discrete tasks might not transfer to rhythmic movements. The results of the endpoint-impedance estimation described in this study also indicate significant variations of the impedance parameters at three key point phases of the ball-bouncing cycle.

The lowest stiffness values are obtained at phase $\varphi_1$, which takes place 0.3 s after the impact and progresses to reach the highest values right before the impact at phase $\varphi_3$, 0.3 s before the impact. As detailed in the result section, this increase is almost linear throughout the three phases, with a significant growth of 53% between $\varphi_1$ and $\varphi_3$, implying smaller displacements when subject to equivalent force.

Counter-intuitively, the endpoint damping reached its lowest values at $\varphi_3$ (no significant difference with $\varphi_1$) and highest value in the middle of the decreasing phase at $\varphi_2$, with

significant differences from the other two phases. One might have expected a behavior correlated with the endpoint stiffness to improve perturbation rejection; in fact, the damping factor of a second-order model is proportional to the damping $B$ of the impedance model, and the settling time of such a model is inversely proportional to the damping factor. Considering the average parameters of all participants for each phase, the settling time in response to a force pulse of both $\varphi_1$ and $\varphi_3$ is 0.43 s, while it is about 0.34 s for $\varphi_2$. Therefore, at phase $\varphi_2$, the arm returns faster to its equilibrium position. Endpoint damping variations seem uncorrelated with the expected requirements imposed by the hybrid task since an increase in the time response would decrease the perturbation rejection performance.

As previously stated, the identifications are conducted on 150 ms; the assumption of constant parameters was also made during that time frame. Even with this assumption, significant variations are observed within 0.6 s. Because of the assumption of constant parameters, parameters estimated at $\varphi_1$ would be shifted away from the perturbation (since they are averaged on 150 ms) while moved closer to the perturbation for $\varphi_3$, making both phases positioned in a positive acceleration cyclic phase, as opposed to $\varphi_2$. Therefore, damping variation could be related to velocity or the sign of acceleration.

The increase in response time to perturbation rejection for the phases close to a predictable perturbation might notify involuntary impedance variation. Indeed, it has been reported that joint impedance is related to joint configuration and velocity [48]. To reject this latter hypothesis, an experience involving similar kinematics but varying interaction force could be tested by changing the ball mass.

## Can endpoint-impedance parameters explain performance?

During a point-to-point task, Burdet et al. [1] reported a selective increase of stiffness in the direction of instability generated by a force field but a low impedance in the direction of the movement. As explained by the authors, since co-contractions have a high metabolic cost and full central nervous system (CNS) control is computationally costly, the CNS is facing an optimization problem to enhance robustness in the direction of perturbation while minimizing metabolic cost. After some trials, the adaptation of the stiffness in the absence of the destabilizing force underlined the learning of the optimal endpoint impedance. Therefore, the CNS can adapt not only the endpoint stiffness magnitude but also the endpoint stiffness geometry (shape and orientation) in a predictive way independent of the force required to compensate for the newly imposed dynamics.

In two other experiments, Erden and Billard [17, 37] demonstrated differences in endpoint impedance. The first experiment compared handedness skills, and the second compared performances between expert and novice participants. They found significant expertise effects on both endpoint stiffness and damping. Novice users tended to have lower average impedance parameters. Similar results were obtained for handedness.

These experiments emphasize the importance of adapting the endpoint-impedance parameters during movements and the relationship between these parameters and expertise or skill. Therefore, it would be expected to find significant differences in endpoint parameters according to performance levels.

However, despite significant performance differences between the three levels chosen, most endpoint-impedance parameters were not significantly different. There might be two main explanations. A first explanation could involve the strategies participants use to improve their performance. Morice et al. [43] and de Rugy et al. [26] reported that the ball-bouncing task might be achieved outside of a passively stable regime thanks to an active control relying on visual information about the ball. Compared to kinematics and timing, our findings could

underline a minor role of viscoelastic endpoint properties in this task. If the impedance is not critical to the task, its variability might increase. The participant's endpoint acceleration at the ball's impacts was on the verge of active and passive control and did not allow conclusions to be drawn. It is also important to stress that no significant impact acceleration variation was found between the different performance levels, suggesting that all participants used passive and active control. To dismiss the hypothesis that impedance variations are not critical to performances in the ball-bouncing task, an experiment with variations of dynamic considerations could be proposed, using the change of the gravity constant or the ball mass.

A second explanation for the endpoint-impedance observations could involve the proposed protocol. Even if some participants had prior experience with the experimental setup, the performance clusters were not defined according to expertise like Erden and Billard [17]; it was instead done retrospectively. Therefore, the clusters chosen, even if significantly different, might not provide important enough differences. The variations caused by performance need to be more critical than individual variations to be statistically significant. To observe slighter variations and reduce noise in the estimated parameters, the position trajectories' precision could be enhanced, or more participants could be included.

## Conclusions

In realistic case studies, joint impedance might be complicated to estimate when human movements are free, and the environment is partially unknown. Endpoint impedance provides a good approximation for slight deviations from a nominal trajectory, allowing the monitoring of global behavior. Our study, in particular, did not discriminate reflexes from intrinsic properties.

The experiments described above revealed significant rapid impedance variations ($<0.6$ s) during the ball-bouncing task, supporting the idea of cyclic variations of the endpoint-impedance during rhythmic tasks. Estimation techniques not relying on the assumption of constant parameters like the ones described by Piovesan et al. [49] or Guarin and Kearney [11] could provide more specific and descriptive cyclic impedance data to both draw more delicate variations and consolidate the findings of this work.

## Supporting information

**S1 Data.**
(PDF)

## Acknowledgments

We would like to thank Baptiste Boyer, Thomas Chevet, Dario Penco, Jeremy Pinguet, and Martin Soyer, who helped with the tuning of the experiment, and the reviewers for their insightful remarks.

## Author Contributions

**Conceptualization:** Isabelle A. Siegler, Pedro Rodriguez-Ayerbe.

**Data curation:** Vincent Fortineau.

**Formal analysis:** Vincent Fortineau.

**Investigation:** Vincent Fortineau, Isabelle A. Siegler, Maria Makarov.

**Methodology:** Vincent Fortineau, Maria Makarov.

**Software:** Vincent Fortineau.

**Supervision:** Isabelle A. Siegler, Maria Makarov, Pedro Rodriguez-Ayerbe.

**Validation:** Isabelle A. Siegler, Maria Makarov, Pedro Rodriguez-Ayerbe.

**Writing – original draft:** Vincent Fortineau.

**Writing – review & editing:** Isabelle A. Siegler, Maria Makarov, Pedro Rodriguez-Ayerbe.

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
