## [Decision Letter · Decision Letter 0]

8 Mar 2023

PONE-D-22-19290Human arm endpoint-impedance in a rhythmic human-robot interaction exhibit cyclic variationsPLOS ONE

Dear Dr. Fortineau,

Thank you for submitting your manuscript to PLOS ONE. After careful consideration, we feel that it has merit but does not fully meet PLOS ONE’s publication criteria as it currently stands. Therefore, we invite you to submit a revised version of the manuscript that addresses the points raised during the review process.

We look forward to receiving your revised manuscript.

Kind regards,

Grzegorz Pochwatko, Ph.D.

Academic Editor

PLOS ONE

Journal Requirements:

Additional Editor Comments (if provided):

Dear Authors,

I am pleased to inform you that your manuscript has been reviewed and assessed by our experts. Based on their recommendations, we would like to invite you to submit a revised version of your manuscript that addresses the minor issues highlighted by the reviewers (both contents and it's form).

I would like to emphasize that your paper has been generally well received. However, there are a few areas that require MINOR REVISIONS to improve the clarity of the manuscript and to ensure that it meets the standards required for publication (proofreading is necessary - important to reach perceptual fluency - very important).

Therefore, I kindly request that you carefully consider and address the comments provided by both reviewers in your revised manuscript. In particular, please focus on the corrections suggested to improve the clarity of your writing (intorduction, discussion, method) and on supplementary materials that would enhance the value of your work. I believe that addressing these comments will significantly strengthen your manuscript.

Please submit the revised version of your manuscript, along with a response to the reviewers' comments (if necessary).

Thank you for your contributions, and I hope to work with you again soon.

Best regards,

gp

Reviewers' comments:

Reviewer's Responses to Questions

**Comments to the Author**

1. Is the manuscript technically sound, and do the data support the conclusions?

Reviewer #1: Yes

Reviewer #2: Yes

2. Has the statistical analysis been performed appropriately and rigorously? 

Reviewer #1: Yes

Reviewer #2: Yes

3. Have the authors made all data underlying the findings in their manuscript fully available?

Reviewer #1: No

Reviewer #2: Yes

4. Is the manuscript presented in an intelligible fashion and written in standard English?

Reviewer #1: Yes

Reviewer #2: Yes

5. Review Comments to the Author

Reviewer #1: Overview: The purpose of this study was to examine variations in end-point impedance of the human arm and hand during physical interactions. In two experiments, 31 participants manipulated digital ball-bouncing with haptic feedback via a robotic plant; the human-robot interaction was monitored to determine endpoint parameters—stiffness, damping, and mass. One experiment correlated impedance with the ball-bouncing height, while the other measured impedance at three different cyclic phases. Ball-bouncing performance was analyzed and clustered to determine if endpoint-impedance affected it. The endpoint-impedance parameters fluctuated rapidly within a cycle but had no direct correlation with ball bounce.

Is the manuscript technically sound, and do the data support the conclusions?

The manuscript is reasonably sound, and the evidence seems to support the conclusions the authors present. However, some technical matters require more attention and discussion.

1. The first part of the Introduction section describes the idea, which I would briefly summarize as “the models should be as simple as they can be but not simpler”. This idea is, however, presented in a convoluted manner. After introducing the utility of “simple mechanical models”, the description delves into specifics and describes “detailed models”. Then, it returns to the “simple models”, narrating specific locomotive CPG-based models afterward. This type of flow is often undesirable. I would suggest following a generally recommended format (often called “the inverted pyramid”) of the Introduction section: (1) introduce essential information, (2) provide important details or supporting information, and (3) list even more specific details.

2. The Introduction section also needs to include the essential details readers want to know without going through references. For example, line 35 refers to “dynamic limbs’ properties” without stating them explicitly; line 37 states “13 parameters” are necessary to account for viscoelastic properties of the muscles but does not hint what the nature of those parameters is.

3. Figures require a more detailed description. For example, although the Figure 1 is self-explanatory, it may be not easy to understand to a first-time reader. I would suggest expanding basic figure descriptions in lines 105, 121, 135, 157, and 223.

4. It is imperative to define abbreviations. In line 32, the abbreviations “KBM” and “KBI” are not previously described. Throughout the text, it becomes clear that K denotes stiffness, B stands for damping, and M is used for mass. However, it is essential that the reader is not forced into guessing and focuses on science rather than deciphering “KBM”, “KBI”, and, perhaps, “ARX” in line 159.

5. The terms of the equation (lines 164-165) are not explained. The description in lines 160-164 could be expanded to clarify how the equation was obtained. The term ∂f (delta f) is ambiguously defined in line 140. Perhaps, the wrong subscript (or the absence of an appropriate superscript) was used to describe the perturbation force.

6. The “Data Analysis” subsection (line 181) describes the use of a k-means algorithm to cluster 31 participants into three performance groups: (1) novice, (2) intermediate, and (3) advanced. It is unclear if the number of groups (clusters) was determined automatically. Could you please explain how you selected the number of groups? In other words, why was the number of groups (centroids in k-means) equal to three and not two? Unless additional custom computations are performed on top, the k-means algorithm runs optimization to distinguish between a predefined number of groups regardless of the dataset you pass. The k-means can also overfit the data. It is therefore necessary to explain the choice of the number of groups.

Has the statistical analysis been performed appropriately and rigorously?

To the best of my knowledge, the statistical analysis been performed appropriately and rigorously. However, minor technical matters may require more attention.

1. The statement about the significant/insignificant results obtained on the same data with Holm/Bonferroni corrections need to be either clarified or removed.

2. Throughout the paper, SD used to abbreviate standard deviation. However, the standard deviations are used inconsistently. For example, the line 257 uses “(6.3 SD)” notation, while the line 258 uses “(SD 4.2)”. Other examples exist throughout the paper.

3. The statements in lines 250-253 may not be clear. The paragraph states that the participants hit the ball at similar vertical positions, listing the ball-bouncing magnitudes that are ~40% different from each other. I might have misunderstood the authors. Clarifications are needed.

Have the authors made all data underlying the findings in their manuscript fully available?

No. The table data are accessible, however the raw data and the computer code for data processing and analysis were not attached.

Is the manuscript presented in an intelligible fashion and written in standard English?

Yes. However, the manuscript requires heavy editing. It has multiple grammar, stylistic, and punctuation errors. For the next submission, I suggest using error correcting software (e.g., Grammarly).

Reviewer #2: This submitted manuscript describes an experimental investigation of the endpoint-impedance of the upper limb while performing a hybrid ball-bouncing task with simulated haptic feedback. A force-perturbation method was used to estimate the endpoint parameters, and statistical measures were calculated for a group of participants. The results indicated that the endpoint-impedance parameters exhibited variations during the arm cycle, but did not show a direct correlation with the performance levels in ball-bouncing.

I find this study interesting; however, several issues should be resolved before the final submission:

1) The paper's aim is presented shortly in the introduction (last paragraph, lines 75-80). It should be elaborated, allowing readers to understand this study's purpose. It is somehow highlighted in the discussion part, where authors compare their results to ones given in the literature.

2) The experimental stand is described briefly in the "Apparatus" subsection. It is accompanied by a simple drawing (Fig. 1). Please, provide a detailed photo of the experimental setup indicating the hardware used.

3) Authors briefly describe the ball-bouncing task. However, the simulation environment needs to be described. Understanding the interaction between the participant, robot, and simulated environment would be crucial. Moreover, it would give better insights into data collection for further analysis.

4) The parameters of equation (1) should be properly described, e.g. is 'k' is a moment of the impact or discrete time? What is the parameter 'alpha'?

5) In Fig. 2, the 'arm position' is given in [cm] and 'ball position' in [m]. It raises the question if the estimated parameters of the participant movement model are correct. Explain if there exists/or not exists the possible scaling effect, as the scale of the perception of the task performed (ball on the screen) and the real task performance (movement of a robot arm) can play a role in the estimation of the parameters. In such a case, the participant's height or body posture can be a significant factor.

6) The ARX identification method should be described in detail.

7) The parameters and the variables in equation (2) are not described.

8) The experiments are indicated with the letters "P" and "H". Please, describe the meaning of these letters.

9) In the Discussion part, the authors write about the results obtained for estimating the KBM model parameters (stiffness, damping, mass). The estimation method must be described properly in the final version of the manuscript.

6. PLOS authors have the option to publish the peer review history of their article (what does this mean?). If published, this will include your full peer review and any attached files.

Reviewer #1: No

Reviewer #2: **Yes: **Jakub Możaryn

---

## [Author Response · Author response to Decision Letter 0]

23 Jun 2023

The response to reviewer have been provided in the rebuttal letter, adressing each point raised.

---

## [Decision Letter · Decision Letter 1]

25 Aug 2023

PONE-D-22-19290R1Human arm endpoint-impedance in rhythmic human-robot interaction exhibit cyclic variationsPLOS ONE

Dear Dr. Fortineau,

Thank you for submitting your manuscript to PLOS ONE. After careful consideration, we feel that it has merit but does not fully meet PLOS ONE’s publication criteria as it currently stands. Therefore, we invite you to submit a revised version of the manuscript that addresses the points raised during the review process.I am inclined towards acceptance.

Required revisions before acceptance (see also 2nd review):

1. Clarify rationale for statistical methods (rationale for both Holm and Bonf.).

2. Address grammatical errors - please give it another round of proofreading.

We look forward to receiving your revised manuscript.

Kind regards,

Grzegorz Pochwatko, Ph.D.

Academic Editor

PLOS ONE

Journal Requirements:

Additional Editor Comments:

I am inclined towards acceptance.

Required revisions before acceptance (see also 2nd review):

1. Clarify rationale for statistical methods (rationale for both Holm and Bonf.).

2. Address grammatical errors - please give it another round of proofreading.

Reviewers' comments:

Reviewer's Responses to Questions

**Comments to the Author**

1. If the authors have adequately addressed your comments raised in a previous round of review and you feel that this manuscript is now acceptable for publication, you may indicate that here to bypass the “Comments to the Author” section, enter your conflict of interest statement in the “Confidential to Editor” section, and submit your "Accept" recommendation.

Reviewer #1: All comments have been addressed

Reviewer #2: All comments have been addressed

2. Is the manuscript technically sound, and do the data support the conclusions?

Reviewer #1: Yes

Reviewer #2: Yes

3. Has the statistical analysis been performed appropriately and rigorously? 

Reviewer #1: No

Reviewer #2: Yes

4. Have the authors made all data underlying the findings in their manuscript fully available?

Reviewer #1: No

Reviewer #2: Yes

5. Is the manuscript presented in an intelligible fashion and written in standard English?

Reviewer #1: Yes

Reviewer #2: Yes

6. Review Comments to the Author

Reviewer #1: Thank you for your efforts during the first round of edits. A few of the issues were addressed; others require extra effort. I encourage you for another round of edits and resubmission. Here, I detail the remaining issues.

Minor issues:

1. The reworked statement about the significant/insignificant results reveals that the authors need to review their statistics again. The description for the use of Holm or Bonferroni corrections does not explain ‘why’ both were used. If the authors want to include the results of both methods in the paper, the rationale for using them should be described. For example, if it is critical to avoid Type I errors (false positives) at all costs, even if it means potentially missing some true effects, then the Bonferroni correction would be appropriate. If you want to balance the trade-off between avoiding false positives and not missing true effects, then Holm's method would be a better choice, as it offers more power while still controlling the family-wise error rate.

2. The data must be accessible for reviewers to validate the truthfulness of the figures.

3. The authors stated a professional reviewed the text. This effort was insufficient. If grammatical errors are difficult to spot, please use free grammar correction software (e.g., Grammarly or others). PLOS ONE guidelines disallow me to pass the manuscript unless it is error-free.

4. There was an improvement in the Introduction, thank you. Still, it remains unclear. The sentences are disconnected and convoluted (the reader cannot follow your argument). The first paragraph of the section, for example, is unreadable. Wordy sentences disservice your paper and should be avoided in favor of clear explanations.

Thank you for this submission and I encourage you to resubmit.

Reviewer #2: The text discusses human endpoint impedance estimation during rhythmic tasks, focusing on a ball-bouncing task. Impedance, including stiffness and damping, helps stabilize movements. During the ball bouncing task, significant cyclic variations in endpoint-impedance parameters were observed. Performance levels, however, may not be directly related to these variations. The study emphasizes the need for more precise estimation techniques and more experiments in order to understand the relationship between endpoint impedance and performance. In addition, there are references to various findings about human arm impedance in different tasks, emphasizing how impedance control impacts motor control. Authors have adequately addressed my comments raised in a previous round of review and in my opinion this manuscript is now acceptable for publication. Language seems clear, correct, and mostly unambiguous based on the text provided. The text's language is well-structured and coherent.

7. PLOS authors have the option to publish the peer review history of their article (what does this mean?). If published, this will include your full peer review and any attached files.

Reviewer #1: No

Reviewer #2: **Yes: **Jakub Możaryn

---

## [Author Response · Author response to Decision Letter 1]

18 Nov 2023

As detailed in the file 'response to reviewer':

- the grammar as been reviewed using AI tools

- the statistical analysis has been done only with Bonferonni (the assumption with Holm analysis have been deleted)

- processed data has been made available on a dataset (https://osf.io/2fapk/?view_only=f1b72ab1413d43178f24c510658ade3e)

- the introduction was reworked

---

## [Editor Report · Decision Letter 2]

28 Nov 2023

Human arm endpoint-impedance in rhythmic human-robot interaction exhibits cyclic variations

PONE-D-22-19290R2

Dear Dr. Fortineau,

We’re pleased to inform you that your manuscript has been judged scientifically suitable for publication and will be formally accepted for publication once it meets all outstanding technical requirements.

Kind regards,

Grzegorz Pochwatko, Ph.D.

Academic Editor

PLOS ONE

Additional Editor Comments (optional):

all reviewers' suggestions have been taken into account, thank you
---

## [Editor Report · Acceptance letter]

6 Dec 2023

PONE-D-22-19290R2 

Human arm endpoint-impedance in rhythmic human-robot interaction exhibits cyclic variations 

Dear Dr. Fortineau:

I'm pleased to inform you that your manuscript has been deemed suitable for publication in PLOS ONE. Congratulations! Your manuscript is now with our production department. 

Kind regards, 

on behalf of

Dr. Grzegorz Pochwatko 

Academic Editor

PLOS ONE